# Impaired Glucose Tolerance and Visceral Adipose Tissue Thickness among Lean and Non-Lean People with and without Spinal Cord Injury

**DOI:** 10.3390/jfmk8030123

**Published:** 2023-08-21

**Authors:** Amy L. Kimball, Michael A. Petrie, Patrick M. McCue, Kristin A. Johnson, Richard K. Shields

**Affiliations:** Department of Physical Therapy and Rehabilitation Science, Carver College of Medicine, The University of Iowa, Iowa City, IA 52242, USA; amy-kimball@uiowa.edu (A.L.K.); michael-petrie@uiowa.edu (M.A.P.); patrick-mccue@uiowa.edu (P.M.M.); kristin-a-johnson@uiowa.edu (K.A.J.)

**Keywords:** metabolic syndrome, glucose metabolism, virtual adiposity, mobility level

## Abstract

After spinal cord injury (SCI), multiple adaptations occur that influence metabolic health and life quality. Prolonged sitting and inactivity predispose people with SCI to body composition changes, such as increased visceral adipose tissue (VAT) thickness, which is often associated with impaired glucose tolerance. Our goal is to understand whether VAT is an index of leanness, and, secondarily, whether mobility methods influence glucose tolerance for people living with SCI. A total of 15 people with SCI and 20 people without SCI had fasting oral glucose tolerance tests (OGTT) and VAT thickness (leanness) measured during a single session. Glucose was 51% and 67% greater for individuals with SCI relative to those without SCI after 60 and 120 min of an OGTT (*p* < 0.001). Glucose area under the curve (AUC) was 28%, 34%, and 60% higher for non-lean people with SCI than lean people with SCI and non-lean and lean people without SCI, respectively (*p* = 0.05, *p* = 0.009, *p* < 0.001). VAT was associated with glucose AUC (R^2^ = 0.23, *p* = 0.004). Taken together, these findings suggest that leanness, as estimated from VAT, may be an important consideration when developing rehabilitation programs to influence metabolism among people with SCI.

## 1. Introduction

Metabolic syndrome affects nearly one in three adults in the United States [1,2], is associated with metabolic syndrome development [3,4], leads to reduced life expectancy [5], and is associated with chronic inflammation [6]. Obesity, a variable that is also associated with metabolic syndrome, is prevalent throughout the US (~30%) [7], but it is nearly two times more prevalent (60%) among people with spinal cord injury (SCI) [8,9]. In this study, we aspire to understand if glucose tolerance is congruent with visceral adipose tissue (VAT) and mode of mobility among people with and without a disability from SCI.

Individuals with SCI undergo a multitude of adaptations that influence glucose tolerance. SCI leads to deleterious adaptations of skeletal, muscular, and nervous system tissues contributing to significant secondary complications, human suffering, and reduced quality of life [10,11,12]. The downstream impact of a lifestyle with minimal activity and secondary environmental factors on human health have prompted widespread attention, including the National Institutes of Health’s “All of Us” research initiative designed to assess molecular and lifestyle biomarkers for people with and without disability [13]. Indeed, we aspire to ultimately understand how lifestyle behaviors, like method of mobility, is associated with VAT and glucose tolerance among people with and without SCI.

People with SCI classically show decreased lean body mass through muscle atrophy [9,14], impaired autonomic nervous system function [15,16], decreased physical activity capacity [17], and transformation of skeletal muscle from an oxidative to a glycolytic phenotype [18]. These adaptations have been linked to glucose intolerance and metabolic syndrome [19,20]. Outwardly, individuals with SCI may not appear to fit the body mass index (BMI) criteria used to classify people for obesity. For example, a BMI > 30 may be rare among people with SCI because of the significant loss of lean skeletal muscle mass from atrophy after paralysis [21,22]. Therefore, we assessed VAT, rather than relying on BMI, to allow us to measure a characteristic of obesity that is independent of lean mass among people with and without SCI. Our primary hypothesis in this study is that people with SCI, who have lower VAT, may have better glucose tolerance than people without a central nervous system disorder who have higher VAT [14,20,23,24,25].

Because of the strong association between glucose tolerance and daily movement [26,27,28], we also sought to assess if the use of manual wheelchairs, which requires daily use of the upper extremities, as compared to powered wheelchairs, which requires minimal use of the upper extremities, would be congruent with glucose tolerance in people with SCI. Many factors influence glucose tolerance [9,29,30,31,32], but prolonged sitting is a known risk for cardio-metabolic disease among people with and without SCI [33,34], while more frequent daily movements during the day are known to mobilize glucose into the muscle with less need for insulin [35]. Accordingly, we secondarily hypothesized that people who used wheeled mobility would show lower levels of VAT and greater glucose tolerance as compared to those who used powered mobility.

The purposes of this study are to (1) compare overall glucose tolerance biomarkers among people with and without SCI; (2) compare glucose tolerance biomarkers in lean and non-lean people with and without SCI; (3) explore the relationship between leanness (VAT) and glucose tolerance in people with and without SCI; and (4), secondarily, assess glucose tolerance levels across methods of mobility for people with and without SCI. We hypothesize that individuals with SCI are more glucose intolerant as compared to people without SCI. We also expect that non-lean individuals with SCI are less glucose tolerant than lean individuals with or without SCI. And finally, we predict that our study will offer support that people with SCI who use powered mobility as compared to wheeled mobility will be less glucose tolerant and have higher VAT levels.

## 2. Materials and Methods

### 2.1. Subjects

Thirty-five subjects were enrolled from the community in the study and were divided into four groups: non-SCI lean (n = 12, 7 female), non-SCI non-lean (n = 8, 3 female), SCI lean (n = 6, 1 female, 2 tetraplegics) and SCI non-lean (n = 9, 1 female, 7 tetraplegics). The mean duration of paralysis for people with SCI was 11.3 (±10.1) in the range of 1 to 30 years. Classification into lean and non-lean groups was based on VAT thickness among people with and without SCI. Ten subjects (4 tetraplegics) utilized a manual wheelchair for mobility while the remaining five subjects with tetraplegia utilized a power wheelchair for mobility. A brief neurologic assessment was performed on all subjects with SCI to confirm lesion level and complete motor SCI (American Spinal Injury Association ASIA-A). Individuals with tetraplegia (C1–C7 lesion) and paraplegia (below C7 lesion) were included in the study. A full health history was not performed and no subjects with or without SCI indicated a previous diagnosis of diabetes mellitus or metabolic syndrome. All subjects provided written informed consent prior to participation that was reviewed by the University of Iowa Human Subjects Office Institutional Review Board and Ethics Committee in compliance with the Declaration of Helsinki.

### 2.2. Visceral Adipose Tissue (VAT) Thickness

VAT was measured using ultrasound imaging (GE LOGIQ e, Fairfield, CT, USA) with the subject in the supine position. A 4C probe was positioned transversely approximately 1 inch superior to the navel at the linea alba to visualize an approximation of the rectus abdominus muscular junction. Probe positioning was adjusted as needed to include the junction of the linea alba, left rectus abdominus, and abdominal aorta. VAT was defined as the distance in centimeters from the anterior abdominal aorta to the junction between the left rectus abdominus and linea alba. The rigor of this assessment method was verified by others [36,37] and by our own assessment of test–retest reliability (ICC 0.989). The threshold of 4 cm thickness was based on our preliminary data and other research that found that in women and men, a visceral fat thickness of 3.55–4.76 cm, respectively, was a cutoff for predicting the presence of metabolic and coronary artery diseases [38].

### 2.3. Oral Glucose Tolerance Testing

Baseline fasting blood glucose was assessed prior to the start of the session using the finger prick method to test capillary glucose levels (Contour^®^ USB Blood Glucose Monitoring System, Bayer HealthCare, Tarrytown, NY, USA). We performed an oral glucose tolerance test using a standard 75 g/10 ounce dextrose solution (Trutol, Thermo Scientific Inc., Waltham, MA, USA). Capillary samples were collected at 6 consistent intervals: baseline, post glucose load, and 30, 60, 90, and 120 min post glucose solution load (Figure 1).

### 2.4. Experimental Protocol

Subjects were tested following an 8 h overnight fast. Prior to the session, baseline anthropometric measurements including weight, height, BMI, and VAT thickness measurements were acquired (Figure 1).

### 2.5. Data Reduction

Descriptive statistics were calculated (mean ± standard deviation) for subject age, height, weight, BMI, and VAT thickness. The glucose area under the curve (AUC) was calculated from the capillary blood glucose levels using the trapezoid method [39].

### 2.6. Statistical Analysis

Statistical analysis was conducted using SigmaStat 11.0 (San Jose, CA, USA). Descriptive statistics were reported as means (±SD). We used independent *t*-tests to determine differences between people with and without SCI for the subject characteristics of age, height, weight, BMI, fasting glucose, and VAT. Additionally, we performed independent *t*-tests to determine differences between lean and non-lean individuals within each population, reported as t(df). We used a mixed model ANOVA to determine differences in glucose concentration during an oral glucose tolerance test between populations (non-SCI versus SCI) and across time (0, 60, and 120 min), where time was repeated within each subject, reported as F(df_1_,df_2_). A one-way ANOVA was used to test for AUC differences among four groups (non-SCI lean, non-SCI non-lean, SCI lean, and SCI non-lean), reported as F(df_1_,df_2_). Additionally, a linear regression analysis was used to calculate coefficients of determination (R^2^) to determine the percent of variance explained by VAT thickness for all participants and within the SCI and non-SCI populations using a significance level of *p* ≤ 0.05. Because of the limited sample size, an exploratory mixed model ANOVA was used to test if participants’ mobility level (ambulatory, manual chair, or powered chair) in some way influenced the glucose concentrations across time (0, 60, and 120 min), where time was repeated within each subject. Lastly, a one-way ANOVA was used to explore differences in VAT thickness across mobility levels. We confirmed all outcome variables passed assumptions of normality using the Shapiro–Wilk procedure. All post hoc testing used the Tukey procedure to test for simple effects across groups, reported as q(k,df). All tests were performed using a significance level of *p* ≤ 0.05.

## 3. Results

### 3.1. Subject Characteristics

Age and BMI significantly differed between the participants with SCI and the participants without SCI (non-SCI) (age *t*(33) = 4.93, *p* < 0.001; BMI *t*(33) = 2.15, *p* = 0.04). Specifically, participants with SCI were older and had a significantly lower BMI compared to participants without SCI. There were no significant differences detected between participants with and without SCI for height, weight, fasting glucose levels, and VAT thickness (Table 1).

For people with SCI, there was a significantly lower age (*t*(13) = −2.33, *p* = 0.04), weight *t*(13) = −3.02, *p* = 0.01), BMI (*t*(13) = −5.00, *p* < 0.001), and VAT thickness (*t*(13) = −9.94, *p* < 0.001) in those classified as SCI lean versus SCI non-lean; however, there was no difference for height and fasting glucose level between the SCI lean and SCI non-lean groups (Table 1).

For people without SCI, there was a significantly lower weight (*t*(18) = −5.26, *p* < 0.001), BMI (*t*(18) = −4.46, *p* < 0.001), fasting glucose (*t*(18) = −2.16, *p* = 0.04), and VAT thickness (*t*(18) = −6.97, *p* < 0.001) in the non-SCI lean group compared to the non-SCI non-lean group; however, there was no difference for height between the non-SCI lean and non-SCI non-lean groups (Table 1).

### 3.2. Glucose Tolerance in SCI versus Non-SCI

There was a significant interaction between the SCI and non-SCI groups across time during the oral glucose tolerance test (F(2,66) = 14.78, *p* < 0.001). Both SCI and non-SCI showed increased glucose levels at 60 min (q(3,66) = 13.74, *p* < 0.001; q(3,66) = 5.97, *p* < 0.001, respectively), while only the SCI group showed a persistent elevation in glucose levels at 120 min compared to baseline (q(3,66) = 10.44, *p* < 0.001). The non-SCI group glucose levels were not significantly higher at 120 min compared to baseline (q(3,66) = 1.62, *p* = 0.49) (Figure 2). The SCI group was significantly higher than the non-SCI group at 60 and 120 min (q(3,66) = 7.304, *p* < 0.001; q(3,66) = 7.67, *p* < 0.001, respectively).

### 3.3. Glucose Tolerance in Lean versus Non-Lean

There was a significant difference in glucose AUC among the four groups (SCI non-lean, SCI lean, non-SCI non-lean, and non-SCI lean) (*F*(3,31) = 10.34, *p* < 0.001; Figure 3). The SCI non-lean glucose (AUC) levels were approximately ~60%, ~34%, and ~28% higher than the non-SCI lean (q(3,31) = 7.82, *p* < 0.001), non-SCI non-lean (q(3,31) = 4.86, *p* = 0.009), and SCI lean (q(3,31) = 3.83, *p* = 0.05), respectively. There were no significant differences in glucose AUC between the SCI lean, non-SCI lean, and non-SCI non-lean groups.

VAT thickness explained approximately ~23% of the variance in glucose AUC for all subjects (R^2^ = 0.23, *p* = 0.004; Figure 4). When each group was analyzed independently, VAT thickness explained approximately ~55% of the variance in glucose AUC in the non-SCI group (R^2^ = 0.55, *p* < 0.001), while approximately ~11% of the variance in glucose AUC was explained by VAT thickness in the SCI group (R^2^ = 0.11, *p* = 0.23).

### 3.4. Glucose Tolerance and Mobility

There was a significant interaction between time and method of mobility for the glucose response (ambulation, power chair, or manual chair) (F(4,64) = 12.59, *p* < 0.001) (Figure 5). Those that could ambulate (non-SCI) had a significantly lower glucose level at 60 and 120 min compared to those who used a manual chair or power chair. Importantly, individuals who used a manual chair had significantly lower glucose levels at 60 and 120 min compared to those who used a power chair (q(4,64) = 5.74, *p* < 0.001, q(4,64) = 6.54, *p* < 0.001, respectively). Additionally, those who used a manual or power chair had a significantly lower glucose level at 120 min compared to 60 min (q(4,64) = 3.512, *p* = 0.041, manual; q(4,64) = 1.29, *p* = 0.64, power).

There were significant differences in the VAT thickness across mobility levels (F(2,32) = 4.80, *p* = 0.015). Importantly, individuals who used a manual wheelchair showed significantly lower VAT thickness compared to power chair users (q(2,32) = 3.57, *p* = 0.043), but not significantly higher than ambulators (q(2,32) = 4.29, *p* = 0.977).

## 4. Discussion

Non-lean individuals with and without SCI have an impaired glucose tolerance that is congruent with increased VAT thickness. People with SCI who use powered wheelchairs for mobility are more likely to have impaired glucose intolerance. The novelty of this study is that we used VAT rather than BMI as our index of leanness to assess people with and without SCI and aggregated the data according to the method used for mobility among people with SCI.

### 4.1. BMI vs. VAT Classification

Our findings support that visceral adiposity may be associated with altered metabolic homeostasis among people with SCI, which is consistent with previous reports [14,30,40]. While others have nicely described factors that regulate glucose metabolism in people with SCI [20,24,40,41], we were able to contribute to this knowledge by showing that BMI was not a meaningful indicator of glucose tolerance in people with SCI. For example, people with SCI showed a lower BMI when compared to our young healthy participants in this study, despite having significant impairment to a glucose challenge and inability to ambulate. A closer examination of other studies supports our findings that BMI poorly predicted obesity and metabolic health [20,30,40,42,43,44,45]. Prior studies have focused on examining the multitude of adaptations that influence health following SCI and comparing SCI to a non-SCI population [45,46]. To our knowledge, this is the first report to specifically examine glucose tolerance in a cohort of SCI and non-SCI individuals and demonstrate that individuals with SCI can present with a phenotypically “healthy” appearance (as supported by lower BMI in conjunction with seemingly normal fasting plasma glucose) yet exhibit altered glucose tolerance. Utilization of VAT thickness as a unique and valid measure allows us to better understand how phenotypic leanness is related to glucose tolerance in people with SCI.

### 4.2. SCI vs. Non-SCI Glucose Response

Physical activity levels and body composition in spinal cord injury are postulated to be an underlying predictor of metabolic disease [6,40,45,47]. Our findings support this, with SCI non-lean persons having higher 2 h oral glucose AUC values compared to lean SCI individuals and all non-SCI individuals (Figure 3). Further, though the SCI individuals in our study had a lower weight, normal fasting glucose levels, and a lower body mass index compared to non-SCI individuals (Table 1), they had significantly elevated glucose levels at both 60 and 120 min (Figure 2), supporting glucose intolerance. This finding is supported by others who found abnormal glucose tolerance in people with SCI [19,29,30,48].

### 4.3. Lean vs. Non-Lean Glucose Response

Elevated glucose AUC may be linked strongly to visceral adipose tissue thickness across all subjects and was especially pronounced in non-lean SCI (Figure 4). In fact, the phenotypically “fit” yet viscerally non-lean individuals with SCI may more closely resemble non-SCI obese individuals regarding glucose intolerance. Our study is unique in examining people with SCI who, by body weight and BMI comparison, appear no different than a non-SCI group yet differ significantly within SCI regarding visceral adiposity. Non-lean responses to an oral glucose challenge were robust when compared to the non-SCI (lean and non-lean) and SCI lean groups. These findings in our study support the notion that visceral adipose composition is associated with glycemic homeostasis in SCI. This is clinically relevant because providers may miss the presence of a pre-diabetic or early diabetic state because the individual appears lean and healthy when using standard assessments such as BMI, body weight, or fasting plasma glucose.

### 4.4. Mobility and Glucose Tolerance in SCI

Our findings support that the method of mobility does influence glucose tolerance in people with SCI. Interestingly, individuals with SCI who were able to self-propel did exhibit improved glucose metabolism compared to those who utilized a power chair (Figure 5). The presence of higher VAT thickness, such as in non-lean people with SCI, leads to less flexibility during glucose metabolism (Figure 5). As expected, SCI with lower VAT thickness and manual chair mobility exhibited improved glucose metabolism. Our finding was supported by Raymond and colleagues who showed that the physical activity level was a strong determinant of 2 h plasma glucose levels in people with SCI [29]. Physical activity is a well-established method to modify cardiometabolic risk factors at a relatively low cost in both non-SCI and SCI individuals [49,50,51,52,53,54]. We recently showed that electrically induced exercise reduced post-meal insulin among people with SCI [19]. Mobility status may serve as a tool to help clinicians identify and monitor patients for the risk of developing abnormal glucose tolerance despite a healthy appearing phenotype, as assessed by BMI. Several participants with SCI that used powered mobility in this study were paraplegic and not tetraplegic. Although we did not have adequate power to statistically stratify this variable, we are aware of a growing trend where all people with paralysis are seeking powered mobility in lieu of wheeled mobility. To our knowledge, our study is the first to include mobility status (manual vs. power chair) in combination with a glucose tolerance test. We acknowledge, however, that there is a complex interaction between activity level and lesion level that needs to be fleshed out in future studies. Other intervention trials support that physical activity, which may be elevated in those using manual wheelchairs, can improve glucose tolerance in SCI [51,52,54,55,56].

### 4.5. Limitations

We did not have adequate power to stratify our groups by age, sex, lesion level, length of injury, socio-economic status, education, and abusive/addictive behaviors, which are all factors that may influence the results of our study. We did not seek a non-SCI group that matched the age of our SCI group because our previous work, and the work of others, supports that age is not a meaningful co-variate of skeletal muscle function among people with SCI [18,19,57,58]. Specifically, young people with SCI portray a skeletal muscle phenotype that is very different from anyone, regardless of age, who has volitional control of their muscles. Importantly, skeletal muscle is deemed a powerful endocrine organ that assists all age groups in reducing post-prandial glucose levels. Hence, our goal was to study people without SCI as an indicator of “innervated” skeletal muscle under volitional control, which includes healthy people who are young and old. We have shown that young people with SCI have a muscle phenotype (nearly 100% fast and fatigable) that is unlike any skeletal muscle of people without SCI, young or old [18,59,60,61,62,63,64]. Nonetheless, we are unable to comment on an age effect on paralyzed skeletal muscle and metabolism within this study.

## 5. Conclusions

The major findings of this study are as follows: (1) people with SCI who are not lean have greater glucose intolerance when compared to lean people with SCI and all people without SCI; (2) visceral fat is a moderate predictor of glucose response among people with and without SCI; and (3) an exploratory study provided preliminary data supporting that mobility status influences glucose regulation among people with SCI; however, the patient case mix was inconsistent across mobility methods. Taken together, these findings suggest that lifestyle factors that influence leanness, as measured by VAT, are important considerations during the rehabilitation of people with SCI. However, more research, with larger sample sizes, is needed to understand the complexities of age, gender, and lesion level on mobility and glucose tolerance among people with SCI.

## Figures and Tables

**Figure 1 jfmk-08-00123-f001:**
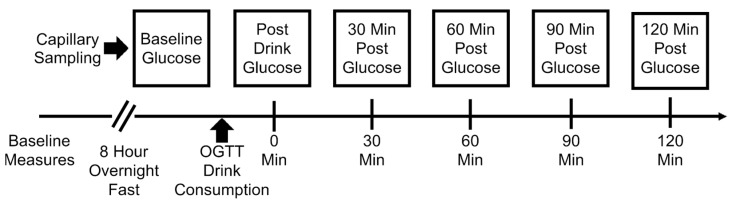
Schematic representation of the study timeline, including baseline assessments and capillary glucose measurements during the oral glucose tolerance test.

**Figure 2 jfmk-08-00123-f002:**
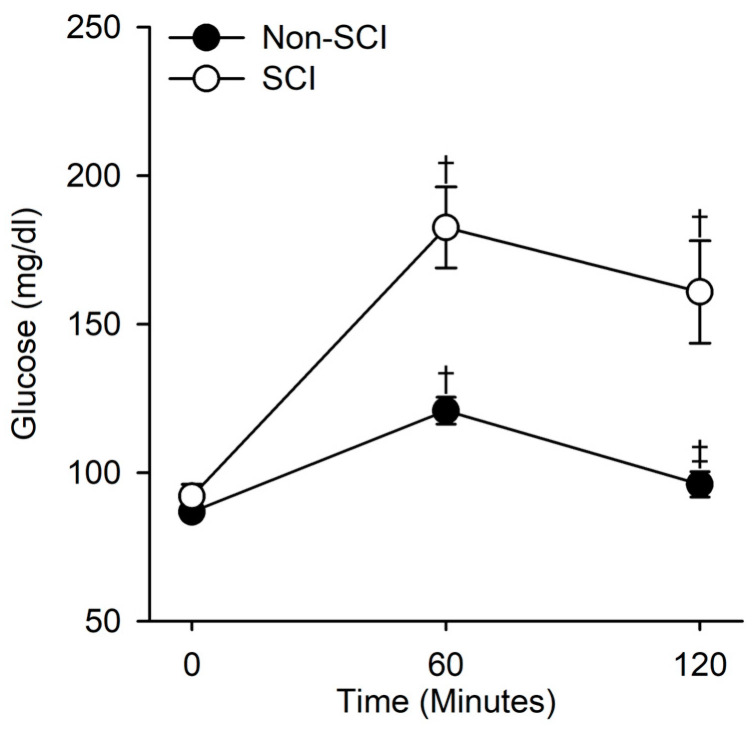
Glucose levels in response to an oral glucose challenge in SCI and non-SCI individuals. There was a significant interaction between population (SCI and non-SCI) and time across the glucose challenge (*p* < 0.001). Within SCI, glucose was approximately 98% and 75% higher at 60 and 120 min compared to baseline (0 min) levels, respectively (*p* < 0.001, both). Within non-SCI, glucose was approximately 40% and 26% higher at 60 min compared to baseline and after 120 min, respectively (*p* < 0.001, *p* = 0.009). Across time, there was no difference at baseline between SCI and non-SCI persons (*p* = 0.66). However, glucose was 51% and 67% greater in SCI than in non-SCI after 60 and 120 min, respectively (*p* < 0.001, both). † indicates a significant difference (*p* < 0.05) at either 60 or 120 min compared to the within-group control at baseline (0 min). ‡ indicates a significant difference (*p* < 0.05) at 120 min compared to the within-group control at 60 min.

**Figure 3 jfmk-08-00123-f003:**
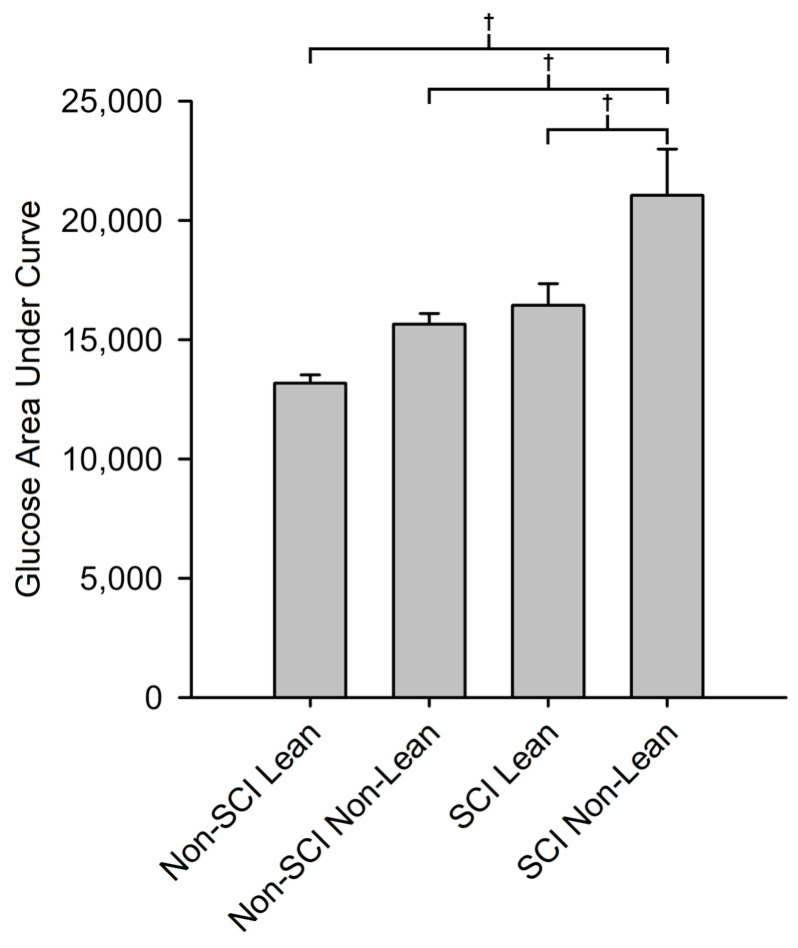
Glucose area under the curve (AUC trapezoid method) for lean and non-lean SCI and non-SCI groups. The AUC for SCI non-lean was 28%, 34%, and 60% higher than the AUC values for SCI lean, non-SCI non-lean, and non-SCI lean, respectively (*p* = 0.05, *p* = 0.009, *p* < 0.001). † indicates a significant difference (*p* < 0.05) between the indicated groups.

**Figure 4 jfmk-08-00123-f004:**
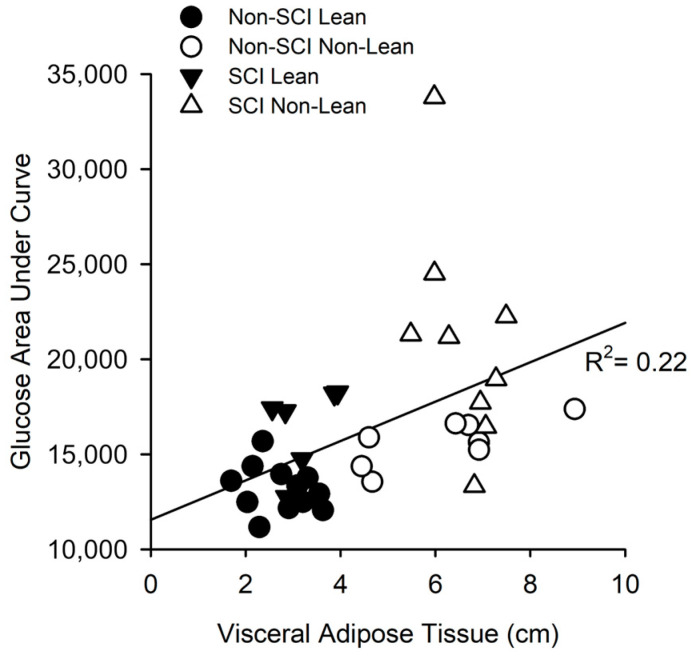
Linear regression analysis of glucose AUC and visceral adipose tissue (VAT) thickness. VAT thickness (cm) was a moderate predictor of glucose AUC explaining approximately 23% of the variance (R^2^ = 0.23, *p* = 0.004) across all subjects. Within each population, VAT thickness explained approximately 55% and 11% of the variance in glucose AUC in non-SCI and SCI individuals, respectively (R^2^ = 0.55, *p* < 0.001; R^2^ = 0.11, *p* = 0.232).

**Figure 5 jfmk-08-00123-f005:**
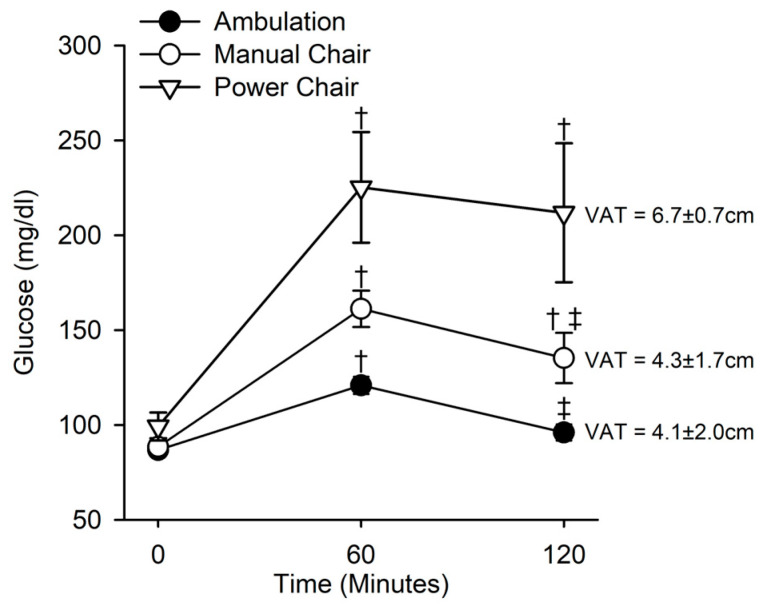
Method of mobility (ambulate, manual wheelchair, or power wheelchair) and the response to an oral glucose tolerance test. There was a significant interaction between the method of mobility and time during the oral glucose tolerance test (*p* < 0.001). While glucose levels for all groups were higher at 60 min compared to baseline (*p* < 0.001, for all), individuals who mobilized using a power wheelchair had the highest glucose levels (*p* < 0.001). Interestingly, those who mobilized using a manual chair had a significantly lower glucose level at 120 min compared to 60 min (*p* < 0.041), whereas those who mobilized using a power chair did not have a significant difference between glucose levels after 60 and 120 min (*p* = 0.64). Further, power wheelchair users had a significantly higher VAT thickness than manual wheelchair users (*p* = 0.04), while the VAT thickness of manual wheelchair users and those that could ambulate (non-SCI) were not significantly different. † indicates a significant difference (*p* < 0.05) at either 60 or 120 min compared to the within-group control at baseline (0 min). ‡ indicates a significant difference (*p* < 0.05) at 120 min compared to the within-group control at 60 min.

**Table 1 jfmk-08-00123-t001:** Subject characteristics during the experimental protocol.

	Non-SCI					SCI				
	All		Lean	Non-Lean		All		Lean	Non-Lean	
	Mean ± SD		Mean ± SD	Mean ± SD		Mean ± SD		Mean ± SD	Mean ± SD	
Subjects	n = 20 (10 females)		n = 12 (7 females)	n = 8 (3 females)		n = 15 (2 females)		n = 6 (1 female)	n = 9 (1 female)	
Age (years)	24.6 ± 2.1	†	24.6 ± 1.4	24.6 ± 2.9		41.3 ± 15.1	†	31.7 ± 8.1	47.8 ± 15.5	‡
Height (cm)	173.7 ± 9.6		172.5 ± 9.8	175.4 ± 9.8		177.6 ± 10.5		178.9 ± 12.4	176.7 ± 9.7	
Weight (kg)	85.3 ± 22.6		71.3 ± 16.3	106.2 ± 11.3	‡	75.8 ± 14.8		64.5 ± 6.08	83.3 ± 14.2	‡
BMI (kg m^−2^)	28.4 ± 7.2	†	24.3 ± 5.8	34.6 ± 3.7	‡	24.0 ± 3.9	†	20.2 ± 1.8	26.5 ± 2.7	‡
Fasting Glucose (mg/dL)	86.8 ± 9.8		83.3 ± 8.4	92.1 ± 9.9	‡	92.1 ± 15.3		90.7 ± 17.6	93.0 ± 14.7	
VAT thickness (cm)	4.1 ± 2.0		2.7 ± 0.6	6.2 ± 1.6	‡	5.3 ± 1.8		3.2 ± 0.6	6.6 ± 0.7	‡

† Significant difference between non-SCI and SCI populations (*p* ≤ 0.05). ‡ Significant difference between lean and non-lean within either respective non-SCI or SCI populations (*p* ≤ 0.05).

## Data Availability

Data are available upon requests made to the corresponding author.

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
