# Peer review of "Impaired Glucose Tolerance and Visceral Adipose Tissue Thickness among Lean and Non-Lean People with and without Spinal Cord Injury"

_jfmk, 2023, doi:10.3390/jfmk8030123_

Round 1

Reviewer 1 Report

The authors examine an important question: do SCI and related attributes illustrate an association with increased glucose intolerance?  The experimental methods and statistics are appropriate.  The presentation is excellent - succinct but with all required context, literature, and analysis.  The conclusions are supported by the data analysis and appropriate post-hoc correction is listed. The discussion places in context the meaning of the results.  The attention to details like the type of mobility (manual chair versus power chair, etc.) really enhance the the utility of the study.’

For completeness, the authors should state whether their data met the normality assumption to complete a t-test.  Use a standard test such as Shapiro-Wilk, etc. If the data does not meet normality assumption, the authors would need to move the analysis to an appropriate method like Kruskal-Wallis.

Otherwise, there was only one minor limitation that should be discussed. Typically, the first 2 years after SCI, most patients have an increased metabolic state after injury often referred to as the “honeymoon period”.  Afterwards, SCI patients tend to gain weight more easily regardless of activity level.  The authors should state the average and standard deviation of time after injury.  Future studies should also consider time since injury as an attribute to examine.

Reviewer 2 Report

On my opinion, authors have done a great job. Undoubtedly, the topic is urgent. The article is well written. A large source material for analysis is presented.

The title is consistent to the problem posed and reflects the main message of the study. However, the determination of visceral adipose tissue thickness (VAT) can be shown in the title, as the authors emphasize this aspect as the main idea. The abstract gives an adequate idea of the entire article.

Introduction

In my opinion, the introduction does not reflect the current state of the issue, of the 26 sources cited, only three relate to the last 5 years. Although the introduction is logical and readable, it is nevertheless necessary to focus on several important aspects, such as: perhaps, in general, about glucose tolerance and the role of visceral fat in changing metabolism, glucose tolerance in SCI and without, focusing on factors that could play a role in the violation of glucose tolerance precisely when SCI. Although the problem of obesity is important, but nevertheless this study has little to do with this aspect, as well as with physical exercise.

The beginning of the introduction Line 23 – there is new data: Hirode G, Wong RJ. Trends in the Prevalence of Metabolic Syndrome in the United States, 2011–2016. JAMA. 2020;323(24):2526–8. or  Liang, X., Or, B., Tsoi, M. F., Cheung, C. L., & Cheung, B. M. Y. (2023). Prevalence of metabolic syndrome in the United States National Health and Nutrition Examination Survey 2011-18. Postgraduate medical journal, qgad008. Advance online publication. https://doi.org/10.1093/postmj/qgad008

Further, the link [2] has little to do with the content, and then there are new data on obesity in the USA, although the link presented [3] is also important / Hales, C.M., Carroll, M.D., Fryar, C.D., & Ogden, C.L. (2017) Prevalence of Obesity Among Adults and Youth: United States, 2015–2016. National Center for Health Statistics Data Brief. Retrieved from https://www.cdc.gov/nchs/data/databriefs/db288.pdf.

There are new data about the US statistics on obesity [4]: Hales, C. M., Carroll, M. D., Fryar, C. D., & Ogden, C. L. (2020). Prevalence of Obesity and Severe Obesity Among Adults: United States, 2017-2018. NCHS data brief, (360), 1–8. or Fan, K., Lv, F., Li, H., Meng, F., Wang, T., & Zhou, Y. (2023). Trends in obesity and severe obesity prevalence in the United States from 1999 to 2018. American journal of human biology: the official journal of the Human Biology Council, 35(5), e23855. https://doi.org/10.1002/ajhb.23855.

The relationship between obesity and life expectancy, for example: Vidra, N., Trias-Llimós, S., & Janssen, F. (2019). Impact of obesity on life expectancy among different European countries: secondary analysis of population-level data over the 1975-2012 period. BMJ open, 9(7), e028086. https://doi.org/10.1136/bmjopen-2018-028086.

There are also reviews on obesity after SCI: Gater, D. R., Jr, Farkas, G. J., & Tiozzo, E. (2021). Pathophysiology of Neurogenic Obesity After Spinal Cord Injury. Topics in spinal cord injury rehabilitation, 27(1), 1–10. https://doi.org/10.46292/sci20-00067.

Vives Alvarado, J. R., Felix, E. R., & Gater, D. R., Jr (2021). Upper Extremity Overuse Injuries and Obesity After Spinal Cord Injury. Topics in spinal cord injury rehabilitation, 27(1), 68–74. https://doi.org/10.46292/sci20-00061

Shojaei, M. H., Alavinia, S. M., & Craven, B. C. (2017). Management of obesity after spinal cord injury: a systematic review. The journal of spinal cord medicine, 40(6), 783–794. https://doi.org/10.1080/10790268.2017.1370207

Sabharwal S. (2019). Addressing cardiometabolic risk in adults with spinal cord injury: acting now despite knowledge gaps. Spinal cord series and cases, 5, 96. https://doi.org/10.1038/s41394-019-0241-5

By visceral fat and SCI: Gorgey, A. S., Ennasr, A. N., Farkas, G. J., & Gater, D. R., Jr (2021). Anthropometric Prediction of Visceral Adiposity in Persons With Spinal Cord Injury. Topics in spinal cord injury rehabilitation, 27(1), 23–35. https://doi.org/10.46292/sci20-00055

Goldsmith, J. A., Ennasr, A. N., Farkas, G. J., Gater, D. R., & Gorgey, A. S. (2021). Role of exercise on visceral adiposity after spinal cord injury: a cardiometabolic risk factor. European journal of applied physiology, 121(8), 2143–2163. https://doi.org/10.1007/s00421-021-04688-3

Edwards, L. A., Bugaresti, J. M., & Buchholz, A. C. (2008). Visceral adipose tissue and the ratio of visceral to subcutaneous adipose tissue are greater in adults with than in those without spinal cord injury, despite matching waist circumferences. The American journal of clinical nutrition, 87(3), 600–607. https://doi.org/10.1093/ajcn/87.3.600

If you look at PUBMED publications, I think you can find a modern representation of the problem.

In my opinion, the purpose of the article is not clearly presented. The purpose of the study should be specified. In this version, the objectives of the study are spelled out.

Materials and Methods

This study meets ethical requirements. The study design is presented, and the methods are clearly explained.  The criteria for selecting study participants are not clearly explained and justified. It is necessary to add a table for patients and healthy people in which to present all anthropometric data. It is necessary to indicate the presence or absence in the anamnesis of metabolic syndrome and/or type 1 or type 2 diabetes in persons with and without SCI. In addition, it is important, before SCI, whether people suffered from metabolic changes, obesity, and to add information for patients about the level of injury, timing, the presence of plegias, the use of a wheelchair type. It is more correct to represent age, height, weight in the form of Me [Min-Max]. For example, 56 [32-62]. From this description of age, it can be seen that patients from 32 to 62 years old participated in the study, and the position of the median indicates a significant shift in the number of patients towards older age. If the authors are aware of the available outliers, it is preferable to use Me [Interquartile Interval], but it is desirable to explain the choice of such descriptive statistics in the text. Statistical significance is denoted as p<0.05 line 123 and 129. Criteria designations are not given. For example, in the results of “F" line 152, “q” and “R"?

Results

When mentioning ratings, in my opinion, it is impossible to name the word "participants" or "antlers". For example, line 132: age and BMI significantly differed between the GCI and-GCI groups........ it should be like this: age and BMI significantly differed between the participants of SCI and non-SCI groups......., and so on in the text.

Section 3.1. is very confusingly written (lines 131-144). It is necessary to describe the participants using a table, starting with a description of the comparison conditions, and not vice versa. Each comparison option should start with a new paragraph. It is probably possible to exclude from the description such parameters as, for example, height, age, since in this case they do not carry important information, but present them in the "methods" section.

Section 3.2 When describing statistical reliability, the entry p<0.001 is used, however, in Fig.2 the figure captions use the entry p<0.05. We need to bring it into line. Usually, one marker uses a comparison based on one attribute (+), another marker based on another attribute (#). Doubling markers means a greater degree of confidence, for example, + - p<0.05; ++ - p<0.001.

In Section 3.2. It is necessary to immediately provide data in the in Lean and Non-lean groups and 4 lines can be displayed in the figure. Since the purpose of the study is to show the difference in the thickness of visceral deposits, it makes no sense to provide separate data for the SCI and Non-SCI groups in general.

How much is Figure 3 needed? Or it may be possible to immediately bring the area under the curve (Glucose area under the curve), and not show the dynamics of changes in glucose levels.

Section 3.4 The title of the subtitle lacks VAT.

I doubt the correctness of comparing a healthy group with patients using a different type of wheelchair. Practically, you compare the same parameters again by combining a group of healthy Lean and Not-lean. How justified is this at all? If you divide this group by the type of activity – maybe it is possible. Therefore, it is necessary to limit ourselves to comparing wheelchair users. And with such a comparison, the question arises – what is the reason for using a chair with manual control and electric drive? Maybe the presence of tetra or paraplegia?

Figure 5 has an incorrect name.

There is no drawing on changing VAT. There is a link to Figure 5, but it refers to the glucose tolerance indicator.

General remark – in the figures, it is necessary to give the number of subjects (n =).

The captions of the figures can be shortened, in my opinion there is no need to repeat the results, you can stop at indicating the trend, and transfer the main text to the results.

Discussion

The first paragraph (lines 222-231) should go to the conclusion.

Further discussion is presented by a very limited amount of literature. Basically, the authors' own results are described, for example, sub-chapters 4.1. and 4.3. which are devoted to the main purpose of the study, do not have a discussion of the results obtained. Again, there is no attraction of modern literature. For example, on the topic [24,31,32]: there are modern reviews that allow us to discuss the results obtained.

Gorgey, A. S., Ennasr, A. N., Farkas, G. J., & Gater, D. R., Jr (2021). Anthropometric Prediction of Visceral Adiposity in Persons With Spinal Cord Injury. Topics in spinal cord injury rehabilitation, 27(1), 23–35. https://doi.org/10.46292/sci20-00055

Goldsmith, J. A., Ennasr, A. N., Farkas, G. J., Gater, D. R., & Gorgey, A. S. (2021). Role of exercise on visceral adiposity after spinal cord injury: a cardiometabolic risk factor. European journal of applied physiology, 121(8), 2143–2163. https://doi.org/10.1007/s00421-021-04688-3

van der Scheer JW, Totosy de Zepetnek JO, Blauwet C, Brooke-Wavell K, Graham-Paulson T, Leonard AN, Webborn N, Goosey-Tolfrey VL. Assessment of body composition in spinal cord injury: A scoping review. PLoS One. 2021 May 7;16(5):e0251142. doi: 10.1371/journal.pone.0251142. PMID: 33961647; PMCID: PMC8104368.

Shin, J. W., Kim, T., Lee, B. S., & Kim, O. (2022). Factors Affecting Metabolic Syndrome in Individuals With Chronic Spinal Cord Injury. Annals of rehabilitation medicine, 46(1), 24–32. https://doi.org/10.5535/arm.21144

Kuvijitsuwan, B., Fongkaew, K., Tengpanitchakul, K., Dolkittanasophon, J., Chunsanit, S., & Pattanakuhar, S. (2022). Correlations between percent body fat measured by dual-energy X-ray absorptiometry and anthropometric measurements in Thai persons with chronic traumatic spinal cord injury. Spinal cord, 60(12), 1094–1099. https://doi.org/10.1038/s41393-022-00828-4

Farkas, G. J., Burton, A. M., McMillan, D. W., Sneij, A., & Gater, D. R., Jr (2022). The Diagnosis and Management of Cardiometabolic Risk and Cardiometabolic Syndrome after Spinal Cord Injury. Journal of personalized medicine, 12(7), 1088. https://doi.org/10.3390/jpm12071088.

I do not agree with the authors that age does not matter when it comes to glucose tolerance: The peak in diabetes prevalence occurs earlier in males than in females and male predominance is specifically observed in middle-aged populations [Tramunt, B., Smati, S., Grandgeorge, N., Lenfant, F., Arnal, J. F., Montagner, A., & Gourdy, P. (2020). Sex differences in metabolic regulation and diabetes susceptibility. Diabetologia, 63(3), 453–461. https://doi.org/10.1007/s00125-019-05040-3].

Glucose tolerance is widely known to deteriorate with age [Swislocki A. L. M. (2022). Glucose Trajectory: More than Changing Glucose Tolerance with Age? Metabolic syndrome and related disorders, 20(6), 313–320. https://doi.org/10.1089/met.2021.0093]

Judging by the results obtained, if not diabetes, then at least prediabetes can be diagnosed in patients using a glucose tolerance test, and diabetes can be diagnosed in patients who use an electric wheelchair. Against this background, it is clear that it was necessary to look at insulin resistance, the condition of the muscles of the lower extremities at least. Or attract similar articles for discussion, rather than articles about physical activity. It is known that the postprandial elevation of plasma glucose stimulates insulin secretion, and the elevation of plasma insulin stimulates glucose uptake in skeletal muscle, leading to the disposal of ingested glucose. Insulin resistance in skeletal muscle begins long before the hyperglycemia becomes evident. In insulin-resistant states, insulin-stimulated glucose uptake in skeletal muscle was markedly impaired. The lower the skeletal muscle mass, the higher the insulin resistance.

 Abdul-Ghani M.A., DeFronzo R.A. Pathogenesis of insulin resistance in skeletal muscle. J. Biomed. Biotechnol. 2010;2010:476279. doi: 10.1155/2010/476279.

Chia, C. W., Egan, J. M., & Ferrucci, L. (2018). Age-Related Changes in Glucose Metabolism, Hyperglycemia, and Cardiovascular Risk. Circulation research, 123(7), 886–904. https://doi.org/10.1161/CIRCRESAHA.118.312806

To discuss the issue of the physical condition of wheelchair patients, it is worth looking for literature on this topic to discuss the results obtained.

For example:

de Groot, S., Valent, L. J., van Koppenhagen, C. F., Broeksteeg, R., Post, M. W., & van der Woude, L. H. (2013). Rolstoelgebruikers met een dwarslaesie in beweging. Effecten van en voorwaarden voor een actieve leefstijl [Physical activity in wheelchair users with spinal cord injury: prerequisites for and effects of an active lifestyle]. Nederlands tijdschrift voor geneeskunde, 157(37), A6220.

van der Scheer, J. W., de Groot, S., Postema, K., Veeger, D. H., & van der Woude, L. H. (2013). Design of a randomized-controlled trial on low-intensity aerobic wheelchair exercise for inactive persons with chronic spinal cord injury. Disability and rehabilitation, 35(13), 1119–1126. https://doi.org/10.3109/09638288.2012.709301

Morse, L. R., Stolzmann, K., Nguyen, H. P., Jain, N. B., Zayac, C., Gagnon, D. R., Tun, C. G., & Garshick, E. (2008). Association between mobility mode and C-reactive protein levels in men with chronic spinal cord injury. Archives of physical medicine and rehabilitation, 89(4), 726–731. https://doi.org/10.1016/j.apmr.2007.09.046

Bass, A., Aubertin-Leheudre, M., Vincent, C., Karelis, A. D., Morin, S. N., McKerral, M., Duclos, C., & Gagnon, D. H. (2020). Effects of an Overground Walking Program With a Robotic Exoskeleton on Long-Term Manual Wheelchair Users With a Chronic Spinal Cord Injury: Protocol for a Self-Controlled Interventional Study. JMIR research protocols, 9(9), e19251. https://doi.org/10.2196/19251

La Fountaine, M. F., Cirnigliaro, C. M., Emmons, R. R., Kirshblum, S. C., Galea, M., Spungen, A. M., & Bauman, W. A. (2015). Lipoprotein heterogeneity in persons with Spinal Cord Injury: a model of prolonged sitting and restricted physical activity. Lipids in health and disease, 14, 81. https://doi.org/10.1186/s12944-015-0084-4

It is surprising that the chapter Limits contains a discussion of his research, which has little to do with this manuscript. In my opinion, it is enough to give a few links, if necessary, of the last years, and not the 90s.

This applies to the entire list of references: out of 44 sources, only 6 come from the last 5 years,

9 are references to their own research.

I believe that the submitted manuscript requires serious work.

Round 2

Reviewer 2 Report

I think that the submitted manuscript has not been corrected. Only technical corrections have been made. The purpose of the study is not clear, so the discussion is a description of the results. My opinion remains the same - the presented article requires semantic processing. So far, I don't see the main hypothesis that the authors are trying to prove.

Round 3

Reviewer 2 Report

I thank the authors for finding the opportunity to make changes to the manuscript. Although in the first response to the reviewer, the authors gave good thoughts that could be included in the discussion or conclusion. I think the manuscript can be accepted for publication.